# GETMusic: Generating Music Tracks with a Unified Representation and Diffusion Framework

## Abstract

Symbolic music generation aims to create musical notes, which can help users compose music, such as generating target instrument tracks based on provided source tracks. In practical scenarios where there's a predefined ensemble of tracks and various composition needs, an efficient and effective generative model that can generate any target tracks based on the other tracks becomes crucial. However, previous efforts have fallen short in addressing this necessity due to limitations in their music representations and models. In this paper, we introduce a framework known as GETMusic, with "GET" standing for "GEnerate music Tracks." This framework encompasses a novel music representation "GETScore" and a diffusion model "GETDiff." GETScore represents musical notes as tokens and organizes tokens in a 2D structure, with tracks stacked vertically and progressing horizontally over time. At a training step, each track of a music piece is randomly selected as either the target or source. The training involves two processes: In the forward process, target tracks are corrupted by masking their tokens, while source tracks remain as the ground truth; in the denoising process, GETDiff is trained to predict the masked target tokens conditioning on the source tracks. Our proposed representation, coupled with the non-autoregressive generative model, empowers GETMusic to generate music with any arbitrary source-target track combinations. Our experiments demonstrate that the versatile GETMusic outperforms prior works proposed for certain specific composition tasks. Readers are invited to listen to our music demos.[1] We promise to open our code.[2]

## 1 Introduction

Symbolic music generation aims to create musical notes, which can help users in music composition. Due to the practical need for flexible and diverse music composition, the need for an efficient and unified approach capable of generating arbitrary tracks based on the others is high.[3] However, current research falls short of meeting this demand due to inherent limitations imposed by their representations and models. Consequently, these approaches are confined to specific source-target combinations, such as generating piano accompaniments based on melodies.

Current research can be categorized into two primary approaches based on music representation: sequence-based and image-based. On one hand, sequence-based works (Huang & Yang, 2020; Zeng et al., 2021; Christopher, 2011) represent music as a sequence of discrete tokens, where a musical note requires multiple tokens to describe attributes such as onset, pitch, duration, and instrument. These tokens are arranged chronologically, resulting in the interleaving of notes from different tracks, and are usually predicted by autoregressive models sequentially. The interleaving of tracks poses a challenge of precise target generation because the autoregressive model implicitly determines when to output a target-track token and avoids generating tokens from other tracks. It also complicates the

---

[1] https://getmusicdemo.github.io/

[2] https://anonymous.4open.science/r/Music1/

[3] A music typically consists of multiple instrument tracks. In this paper, given a predefined track ensemble, we refer to the tracks to be generated as "target tracks" and those acting as conditions as "source tracks." We refer to such an orchestration of tracks as a "source-target combination."

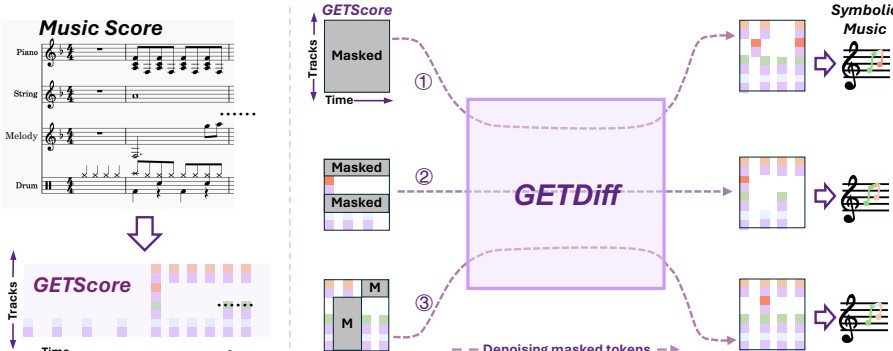

Figure 1: The overview of GETMusic, involving a novel music representation "GETScore" and a discrete diffusion model "GETDiff." Given a predefined ensemble of instrument tracks, GETScore represents the music by encoding musical notes into discrete tokens and arranging them in a two-dimensional format that is both temporally and track-specific. GETDiff takes GETScores as inputs and generates any desired target tracks from scratch (①) or conditioning on any source tracks (②). This flexibility extends beyond track-wise generation, as it can perform zero-shot generation for any irregular masked parts (③).

specification of source and target tracks. Therefore, the existing methods (Dong et al., 2023; Ren et al., 2020; Yu et al., 2022) typically focus on either one specific source-target track combination or the continuation of tracks.

On the other hand, image-based research represents music as 2D images, with pianorolls[4] being a popular choice. Pianorolls represent musical notes as horizontal lines, with the vertical position denoting pitch and the length signifying duration. A pianoroll explicitly separates tracks but it has to incorporate the entire pitch range of instruments, resulting in large and sparse images. Due to the challenges of generating sparse and high-resolution images, most research has focused on conditional composition involving only a single source or target track (Dong et al., 2017; Yang et al., 2017; Shuyu & Sung, 2023) or unconditional generation (Mittal et al., 2021).

To support the generation across flexible and diverse source-target track combinations, we propose a unified representation and diffusion framework called GETMusic ("GET" stands for **GE**nerate music **T**racks), which comprises a representation named GETScore, and a discrete diffusion model (Austin et al., 2021) named GETDiff. GETScore represents the music as a 2D structure, where tracks are stacked vertically and progress horizontally over time. Within each track, we efficiently represent musical notes with the same onset by a single pitch token and a single duration token, and position them based on the onset time. At a training step, each track in a training sample is randomly selected as either the target or the source. The training consists of two processes: In the forward process, the target tracks are corrupted by masking tokens, while the source tracks are preserved as ground truth; in the denoising process, GETDiff learns to predict the masked target tokens based on the provided source. Our co-designed representation and diffusion model in GETMusic offer several advantages compared to prior works:

- With separate and temporally aligned tracks in GETScore, coupled with a non-autoregressive generative model, GETMusic adeptly compose music across various source-target combinations.

- GETScore is a compact multi-track music representation while effectively preserving interdependencies among simultaneous notes both within and across tracks, fostering harmonious music generation.

- Beyond track-wise generation, the mask and denoising mechanism of GETDiff enable the zero-shot generation (i.e., denoising masked tokens at any arbitrary locations in GETScore), further enhancing the versatility and creativity.

We demonstrate that our proposed versatile GETMusic surpasses approaches proposed for specific tasks such as conditional accompaniment or melody generation, as well as generation from scratch.

---

[4]https://en.wikipedia.org/wiki/Piano_roll

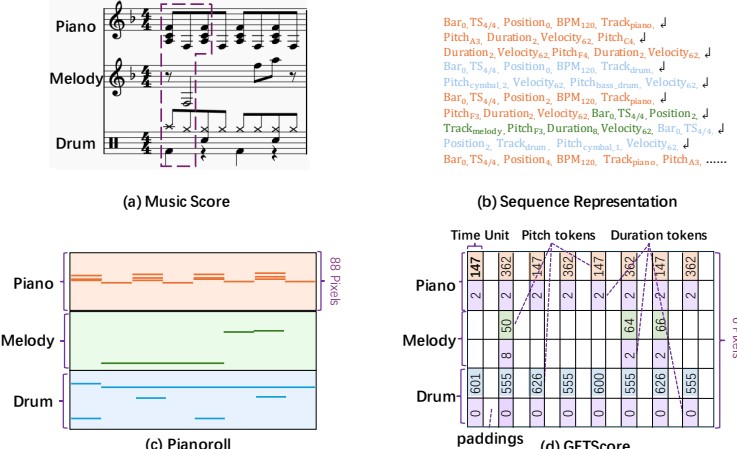

Figure 2: Different representations for the same piece of music. Figure (a) is the music score. Figure (b) illustrates the sequence-based representation in REMI (Huang & Yang, 2020) style, and due to the length of the sequence, we only show the portion enclosed by the dashed box in Figure (a). Figure (c) shows a sparse pianoroll that represents notes by lines. In Figure (d), GETScore separates and aligns tracks, forming the basis for unifying generation across various source-target combinations. It efficiently preserves the interdependencies among simultaneous notes, fostering generation harmony. Numbers in (d) denote token indices.

## 2 BACKGROUND

### 2.1 SYMBOLIC MUSIC GENERATION

Symbolic music generation aims to generate musical notes, whether from scratch (Mittal et al., 2021; Yu et al., 2022) or based on given conditions such as chords, tracks (Shuyu & Sung, 2023; Huang & Yang, 2020; Dong et al., 2017), lyrics (Lv et al., 2022; Ju et al., 2021; Sheng et al., 2020), or other musical properties (Zhang et al., 2022), which can assist users in composing music. In practical music composition, a common user need is to create instrumental tracks from scratch or conditioning on existing ones. Given a predefined ensemble of tracks and considering flexible composition needs in practice, a generative model capable of handling arbitrary source-target combination is crucial. However, neither of the existing approaches can integrate generation across multiple source-target combinations, primarily due to inherent limitations in their representations and models.

Current approaches can be broadly categorized into two main categories with respect to adopted representation: sequence-based and image-based. In sequence-based methods (Huang & Yang, 2020; Hsiao et al., 2021; Zeng et al., 2021; Ren et al., 2020), music is represented as a sequence of discrete tokens. A token corresponds to a specific attribute of a musical note, such as onset (the beginning time of a note), pitch (note frequency), duration, and instrument, and tokens are usually arranged chronologically. Consequently, notes that represent different tracks usually interleave, as shown in Figure 2(b) where the tracks are differentiated by colors. Typically, an autoregressive model is applied to processes the sequence, predicting tokens one by one. The interwove tracks and the autoregressive generation force the model to implicitly determine when to output tokens of desired target tracks and avoid incorporating tokens belonging to other tracks, which poses a challenge to the precise generation of the desired tracks; the sequential representation and modeling do not explicitly preserve the interdependencies among simultaneous notes, which impact the harmony of the generated music; furthermore, the model is required to be highly capable of learning long-term dependencies (Bengio et al., 1994) given the lengthy sequences. Some unconventional methods (Ens & Pasquier, 2020) organize tokens according to the track order in order to eliminate track interleaving. However, it comes with a trade-off, as it results in weaker dependencies both in the long term and across tracks.

Image-based methods mainly employ pianoroll representations which depict notes as horizontal lines in 2D images, with the vertical position denoting pitch and the length signifying duration. However, pianorolls need to include the entire pitch range of the instrument, resulting in images that are both large and sparse. For instance, Figure 2(c) illustrates a pianoroll representation of a three-track music piece, which spans a width of hundreds of pixels, yet only the bold lines within it carry musical

information. Most works focus on conditional composition involving only a single source/target track (Dong et al., 2017; Yang et al., 2017; Shuyu & Sung, 2023) or unconditional generation (Mittal et al., 2021) because generating a sparse and high-resolution image is challenging.

Our proposed GETMusic addresses above limitations with a co-designed representation and a discrete diffusion model which together provide an effective solution to versatile track generation.

## 2.2 DIFFUSION MODELS

Diffusion models, initially proposed by (Sohl-Dickstein et al., 2015) and further improved by subsequent research (Ho et al., 2020; Song et al., 2021; Ho & Salimans, 2021; Dhariwal & Nichol, 2021), have demonstrated impressive capabilities in modeling complex distributions. These models consist of two key processes: a forward (diffusion) process and a reverse (denoising) process. The forward process $q(x_{1:T}|x_0) = \prod_{t=1}^{T} q(x_t|x_{t-1})$ introduces noise to the original data $x_0$ iteratively for $T$ steps, corrupting it towards a prior distribution $p(x_T)$ that is independent of $x_0$. The goal of diffusion models is to learn a reverse process $p_\theta(x_{t-1}|x_t)$ that gradually denoises $x_T$ to the data distribution. The model is trained by optimizing the variational lower bound (VLB) (Ho et al., 2020):

$$L_{\text{vlb}} = \mathbb{E}_q[-\log p_\theta(x_0|x_1)] + D_{KL}[q(x_T|x_0)||p(x_T)]]$$
$$+ \sum_{t=2}^{T} D_{KL}[q(x_{t-1}|x_t,x_0)||p_\theta(x_{t-1}|x_t))] . \tag{1}$$

The models that operate within the latent space are commonly referred to as continuous diffusion models, whereas the discrete diffusion models (Sohl-Dickstein et al., 2015) were developed for binary sequence learning. (Hoogeboom et al., 2021) extended these models to handle categorical random variables, while D3PM (Austin et al., 2021) introduced a more structured categorical forward process: the forward process is a Markov chain defined by transition matrices, which transitions a token at time $t-1$ to another at time $t$ by probability. In the reverse process, a pivotal technique known as $x_0$-parameterization was introduced by (Austin et al., 2021). Instead of directly predicting $x_{t-1}$ at time step $t$, the model learns to fit the noiseless original data $x_0$ and corrupts the predicted $\tilde{x}_0$ to obtain $x_{t-1}$. Consequently, an auxiliary term scaled by a hyper-parameter $\lambda$ is added to the VLB:

$$L_\lambda = L_{\text{vlb}} + \lambda \mathbb{E}_q \left[ \sum_{t=2}^{T} -\log p_\theta(x_0|x_t) \right] . \tag{2}$$

Mittal et al. (2021) first applied continuous diffusion models to music generation. However, due to technical limitations at the time, their approach is not fully end-to-end and is restricted to generating single-track music unconditionally. They opted for a pianoroll representation in their model. SDMuse (Zhang et al., 2022) considered the limitations inherent in the pianoroll and sequence representation. They first employ a continuous diffusion model that operates on pianoroll, followed by an autoregressive model to refine the music sequence converted from generated pianorolls. This two-stage pipeline significantly increases the computational cost but it does not fundamentally address the performance bottleneck, i.e., accurately generating sparse and high-resolution pianorolls.

In contrast to these prior works, our GETDiff is an end-to-end discrete diffusion model. The rationale behind the discrete model lies in the fact that GETDiff is specifically crafted to operate on GETScore, which comprises discrete tokens, aligning perfectly with the inherently discrete nature of symbolic music. Moreover, recent studies (Gu et al., 2022; Tang et al., 2022) have demonstrated that, for conditional generation, discrete diffusion models offer many advantages over continuous diffusion models. Meanwhile, our novel representation combines the strengths of conventional representations while circumventing their shortcomings, thereby overcoming the obstacles associated with generating high-quality content.

## 3 GETMUSIC

In this section, we introduce two key components in GETMusic: the representation GETScore and the diffusion model GETDiff. We first provide an overview of each component, and then highlight their advantages in supporting the flexible and diverse generation of any tracks.

Figure 3: An overview of training the GETDiff using a 3-track GETScore. Note that GETScore is capable of accommodating any number of tracks, with this example serving as a toy example. During this training step, the piano track is randomly selected as the source and the drum track as the target, while the melody track is ignored. Thus, $x_t$ consists of the ground truth piano track, an emptied melody track, and a corrupted drum track. GETDiff generates all tokens simultaneously in a non-autoregressive manner which may modify tokens in its output. Therefore, when $x_{t-1}$ is obtained, the sources are recovered with the ground truth while ignored tracks are emptied again.

## 3.1 GETSCORE

Our goal is to design an efficient and effective representation for modeling multi-track music, which allows for flexible specification of source and target tracks and thereby laying the foundation of the diverse track generation tasks. Our novel representation GETScore involves two core ideas: (1) the 2D track arrangement and (2) the musical note tokenization.

**Track arrangement**    We derive inspiration from music scores to arrange tracks vertically, with each track progressing horizontally over time. The horizontal axis is divided into fine-grained temporal units, with each unit equivalent to the duration of a 16th note. This level of temporal detail is sufficient to the majority of our training data. This arrangement of tracks brings several benefits:

• It prevents content of different tracks from interleaving, which simplifies the specification of source and target tracks, and facilitates the precise generation of desired tracks.

• Because tracks are temporally aligned like music scores, their interdependencies are well preserved.

**Note tokenization**    To represent musical notes, we focus on two attributes: pitch and duration, which are directly associated with composition. Some dynamic factors like velocity and tempo variation fall outside the scope of our study. We use two distinct tokens to denote a note's pitch and duration, respectively. These paired pitch-duration tokens are placed in accordance with the onset time and track within GETScore. Some positions within GETScore may remain unoccupied by any tokens; in such instances, we employ padding tokens to fill them, as illustrated by the blank blocks in Figure 2(d). Each track has its own pitch token vocabulary but shares a common duration vocabulary, considering pitch characteristics are instrument-dependent, whereas duration is a universal feature across all tracks. To broaden the applicability of GETScore, we need to address two more problems:

(1) How to use single pitch and duration tokens to represent a group of notes played simultaneously *within a track*? We propose merging pitch tokens of a group of simultaneous notes into a single compound pitch token. Furthermore, we identify the most frequently occurring duration token within the group as the final duration token. This simplification of duration representation is supported by our observation from the entire training data, in only 0.5% groups, the maximum duration difference among notes exceeds a temporal unit. These findings suggest that this simplification has minimal impact on the expressive quality of GETScore. Figure 2(d) illustrates the compound token: in the piano track, we combine three simultaneous 1/8 notes played at the first beat, namely, "La", "Do", and "Fa," into a single pitch token indexed with "147" alongside a duration token "2."

(2) How to represent percussive instruments, such as drums, which do not involve the concepts of "pitch" and "duration?" We treat individual drum actions (e.g., kick, snare, hats, toms, and cymbals)

as pitch tokens and align them with a special duration token. The drum track in Figure 2(d) illustrates our approach.

In conclusion, besides the benefits from track arrangement, our GETScore also gains advantages through this note tokenization:

• Each track requires only two rows to accommodate the pitch and duration tokens, significantly enhancing the efficiency of GETScore.

• The compound token preserves the interdependecies within a track. When a token is generated, the harmony is guaranteed because the corresponding note group is derived from real-world data.

## 3.2 GETDIFF

We first introduce the forward and the denoising process of GETDiff. Next, we introduce the inference procedure and outline GETDiff's benefits for diverse generation needs.

**The forward process**  Our discrete diffusion model GETDiff takes GETScores as inputs. We introduce a special token [MASK] into the vocabulary as the absorbing state of the forward process. At time $t - 1$, a normal token remains in its current state with a probability of $\alpha_t$ and transitions to [MASK] (i.e., corrupts to noise) with a probability of $\gamma_t = 1 - \alpha_t$. As GETScore includes a fixed number of tracks that GETMusic supports, and the composition does not always involve all tracks, we fill the uninvolved tracks with another special token [EMPTY]. [EMPTY] never transitions to other tokens, nor can it be transitioned to from any other tokens. This design prevents any interference from uninvolved tracks in certain compositions. Formally, a transition matrix $[Q_t]_{mn} = q(x_t = m | x_{t-1} = n) \in \mathbb{R}^{K \times K}$ defines the transition probability from the $n$-th token at time $t - 1$ to the $m$-th token at time $t$:

$$Q_t = \begin{bmatrix} \alpha_t & 0 & \dots & 0 & 0 \\ 0 & \alpha_t & \dots & 0 & 0 \\ \vdots & \vdots & \ddots & \vdots & \vdots \\ 0 & 0 & \dots & 1 & 0 \\ \gamma_t & \gamma_t & \dots & 0 & 1 \end{bmatrix}, \tag{3}$$

where $K$ is the total vocabulary size, including two special tokens. The last two columns of $Q_t$ correspond to the probability $q(x_t | x_{t-1} = \text{[EMPTY]})$ and $q(x_t | x_{t-1} = \text{[MASK]})$, respectively. Denoting $v(x)$ as a one-hot column vector indicating the category of $x$ and considering the Markovian nature of the forward process, we can express the marginal at $t$, and the posterior at $t - 1$ as:

$$q(x_t | x_0) = v^\top(x_t) \overline{Q}_t v(x_0), \quad \text{with} \quad \overline{Q}_t = Q_t \dots Q_1. \tag{4}$$

$$q(x_{t-1} | x_t, x_0) = \frac{q(x_t | x_{t-1}, x_0) q(x_{t-1} | x_0)}{q(x_t | x_0)} = \frac{\left(v^\top(x_t) Q_t v(x_{t-1})\right) \left(v^\top(x_{t-1}) \overline{Q}_{t-1} v(x_0)\right)}{v^\top(x_t) \overline{Q}_t v(x_0)}. \tag{5}$$

**The denoising process**  Figure 3 provides an overview of GETDiff denoising a three-track training sample of a length of $L$ time units. GETDiff has three main components: an embedding module, Roformer (Su et al., 2021) layers, and a decoding module. Roformer is a Transformer (Vaswani et al., 2017) variant that incorporates relative position information into the attention matrix, which enhances the model's ability to length extrapolation during inference.

During training, GETDiff needs to cover the various source-target combinations for a music piece with $I$ tracks, represented as a GETScore with $2I$ rows. To achieve this, $m$ tracks (resulting in $2m$ rows in GETScore) are randomly chosen as the source, while $n$ tracks (resulting in $2n$ rows in GETScore) are selected as the target, $m \geq 0$, $n > 0$, and $m + n \leq I$.

At a randomly sampled time $t$, to obtain $x_t$ from the original GETScore $x_0$, tokens in target tracks are transitioned according to $\overline{Q}_t$, tokens in the source tracks remain as the ground truth, and uninvolved tracks are emptied. GETDiff denoises $x_t$ in four steps, as shown in Figure 3:

(1) All tokens in GETScore are embedded into $d$-dimensional embeddings, forming an embedding matrix of size $2Id \times L$.

(2) Two types of learnable embeddings, named condition flags, are added to the matrix. These flags differentiate whether a token originates from the provided source ("True") or is generated by the

model ("False"). The rationale behind this design is as follows: GETScore is designed to mimic a music score where each token, including paddings that signify rests or cadences, carries information. Because the tokens predicted at the current time step can act as conditions in the subsequent time step, any inaccuracies in these tokens can lead to deviations from the intended denoising direction during inference, especially in the first few steps. Condition flags explicitly indicates the reliability of the tokens upon which the model is conditioned, thereby enhancing the quality of generation.

(3) The embedding matrix is resized to GETDiff's input dimension $d_{model}$ using an MLP, and then fed into the Roformer model.

(4) The output matrix passes through a classification head to obtain the token distribution over the vocabulary of size $K$ and we obtain the final tokens using the gumbel-softmax technique.

GETDiff employs the $x_0$-parameterization (Austin et al., 2021) (see §2.2) and thus the training objective is defined as Eq.2. The posterior computation for the $L_{vlb}$ term in Eq.2 is provided by Eq.5.

**Inference**   During inference, users can specify any target and source tracks. The resulting GETScore representation is then constructed accordingly, denoted as $x_T$, which contains the ground truth of source tracks, masked target tracks, and emptied tracks (if any). GETDiff then denoises $x_T$ step by step to obtain $x_0$. As GETDiff generates all tokens simultaneously in a non-autoregressive manner, potentially modifying source tokens in the output, we need to ensure the consistent guidance from source tracks: when $x_{t-1}$ is acquired, tokens in source tracks are recovered to their ground truth values, while tokens in uninvolved tracks are once again emptied.

Considering the combined benefits of the representation and the diffusion model, our GETMusic framework offers several major advantages in addressing the diverse composition needs:

• Through a unified diffusion model, GETMusic has the capability to compose music across a range of source-target combinations without requiring re-training.

• Beyond the track-wise generation, the mask and denoising mechanism of GETDiff enables the zero-shot generation of any arbitrary masked locations in GETScore, which further enhances versatility and creativity. An illustration of this can be found in case ③ in Figure 1.

## 4 EXPERIMENTS

### 4.1 EXPERIMENT SETTINGS

**Data**   Due to the data-hungry nature of the diffusion model, we ensured adequate data preparation for its training and testing. We gathered 1,569,469 MIDI files from Musescore.[5] We processed the crawled data basically following the approach outlined in (Ren et al., 2020), resulting in symbolic music data comprising $I = 6$ instrumental tracks: *bass, drum, guitar, piano, string, melody*, along with an additional chord progression track. Further details are provided in Appendix A. After implementing rigorous cleansing and filtering procedures, we produced approximately 140,000 high-quality GETScores, equivalent to approximately 3,000 hours, with the maximum $L$ as 512. We sampled 1,000 GETScores for validation, 1,000 for testing, and the rest for training.

**Tasks and baselines**   We consider three symbolic music generation tasks: (1) *accompaniment generation based on the melody*, (2) *melody generation based on the accompaniments*, and (3) *generating tracks from scratch*.

For Task 1 and 2, we compare GETMusic with PopMAG (Ren et al., 2020), which is an autoregressive Transformer encoder-decoder model that processes a sequence representation MuMIDI. To be comparable, we restrict the generated music to a maximum length of 128 beats, which is the longest composition length for PopMAG. Both PopMAG and GETMusic always employ the ground truth chord progression as a condition (i.e., a source track) for generation in these tasks. For the third task, we compare GETMusic with Museformer (Yu et al., 2022), one of the most competitive unconditional music generation models. For fair comparison, we train all baselines on our crawled data. Task 1 and 2 are evaluated on the test set, while Task 3 is evaluated by unconditionally generating 1,000 pieces of music using different random seeds.

---

[5] https://musescore.com/

**Training details**  We set diffusion timesteps $T = 100$ and the auxiliary loss scale $\lambda = 0.001$. For the transition matrix $Q_t$, we linearly increase $\overline{\gamma}_t$ (cumulative $\gamma_t$) from 0 to 1 and decrease $\overline{\alpha}_t$ from 1 to 0. GETDiff has 12 Roformer layers with $d = 96$ and $d_{model} = 768$, where there are about 86M trainable parameters. During training, we use AdamW optimizer with a learning rate of $1e - 4$, $\beta_1 = 0.9$, $\beta_2 = 0.999$. The learning rate warmups first 1000 steps and then linearly decays. The training is conducted on $8 \times 32G$ Nvidia V100 GPUs and the batch size on each GPU is 3. We train the model for 50 epochs and validate it every 1000 steps. Checkpoints are selected based on the validation loss.

## 4.2  EVALUATION METRICS

**Objective evaluation**  We introduce objective metrics that quantitatively evaluates the generation quality. Following Ren et al. (2020), we evaluate the models from two aspects:

(1) **Chord accuracy**: For Task 1 and 2, we measure the chord accuracy $CA$ between generated target tracks and their ground truth to evaluate the melodic coherence:

$$CA = \frac{1}{N_{tracks} \times N_{chords}} \sum_{i=1}^{N_{tracks}} \sum_{j=1}^{N_{chords}} \mathbb{1}(C_{i,j}^{'} = C_{i,j}). \tag{6}$$

Here, $N_{tracks}$ and $N_{chords}$ represent the number of tracks and chords, respectively. $C_{i,j}^{'}$ and $C_{i,j}$ denote the $j$-th chord in the $i$-th generated target track and the ground truth, respectively. Note that this metric is not suitable for the third task. Instead, melodic evaluation for the third task relies on both the pitch distribution and human evaluation, which are discussed later.

(2) **Feature distribution divergence**: For the first two tasks, we assess the distributions of important musical features in generated and ground truth tracks: note pitch, duration (*Dur*) and Inter-Onset Interval (*IOI*) that measures the temporal interval between two consecutive notes within a bar. First, we quantize the duration and *IOI* into 16 classes, then convert the histograms into probability density functions (PDFs) using Gaussian kernel density estimation. Finally, we compute the KL-divergence (Kullback & Leibler, 1951) $KL_{\{Pitch,Dur,IOI\}}$ between PDFs of generated target tracks and ground truth. For task 3, we compute $KL_{\{Pitch,Dur,IOI\}}$ between PDFs of generated target tracks and the corresponding distribution of training data. In this definition, $KL_{Pitch}$ reflects the melody consistency, whereas $KL_{Dur}$ and $KL_{IOI}$ are indicative of rhythm consistency.

**Human evaluation**  30 evaluators, who possessed a basic knowledge of music, were divided into three groups, each comprising 10 members. Each group took part in a blind test dedicated to one of the three tasks, evaluating all music created by GETMusic and baseline models in this task. Evaluation metrics include "melodic," "rhythmic," and "musical structure," etc., as outlined in Appendix B.

## 4.3  GENERATION RESULTS

**Comparison with previous SOTA methods**  Table 1 presents the objective evaluation results of three tasks. In Task 1 and 2, GETMusic significantly outperforms PopMAG across all metrics with a p-value of less than 0.01 in the t-test. This highlights its ability to create music with more harmonious melodies (higher $CA$ and lower $KL_{Pitch}$) and more suitable rhythms (lower $KL_{Dur}$ and $KL_{IOI}$) that align well with the provided source tracks. In Task 2, where all five accompaniment instruments serve as source tracks, we achieve better scores in all metrics compared to the first task which relies solely on the melody as the source track. This improvement caused by more generation conditions aligns with intuitive expectations. In Task 3, GETMusic outperforms the competitive baseline Museformer in most metrics. Subjective evaluations further confirm our effectiveness (Figure 4): In every aspect, GETMusic's human ratings outperform the competitive baselines. $\kappa$ values for three tasks are 0.69, 0.72, and 0.62, all indicating a substantial agreement among the evaluators.

**Zero-shot generation**  Although GETMusic is trained for track-wise generation, it can zero-shot recover masked tokens at any arbitrary locations, due to its the mask and denoising mechanism. The zero-shot generation is examplified in case ③ in Figure 1. This capability enhances the versatility and creativity of GETMusic. For example, we can insert mask tokens in the middle of two different songs to connect them: GETMusic generates a harmonious bridge by iteratively denoising the masked tokens while preserving the rest of the tokens unchanged. Despite the challenges in evaluation, the 7th and 8th demos on the demo page showcase our approach's flexibility and creativity.

Table 1: Objective evaluation on three representative tasks: the accompaniment/melody generation as well as generating from scratch. In the third task, where ground truth chord progressions are absent, $CA$ evaluation is unavailable.

| Method | $CA(\%)\uparrow$ | $KL_{Pitch}\downarrow$ | $KL_{Dur}\downarrow$ | $KL_{IOI}\downarrow$ |
|---|---|---|---|---|
| Task 1: Accompaniment Generation | | | | |
| PopMAG | 61.17 | 10.98 | 7.00 | 6.92 |
| **GETMusic** | **65.48** | **10.05** | **4.21** | **4.22** |
| Task 2: Lead Melody Generation | | | | |
| PopMAG | 73.70 | 10.64 | 3.97 | 4.03 |
| **GETMusic** | **81.88** | **9.82** | **3.67** | **3.49** |
| Task 3: Generation from Scratch | | | | |
| Museformer | - | 8.19 | **3.34** | 5.71 |
| **GETMusic** | - | **7.99** | 3.38 | **5.33** |

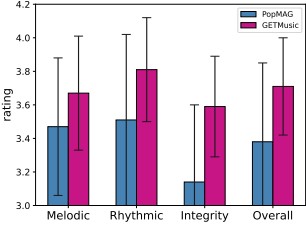

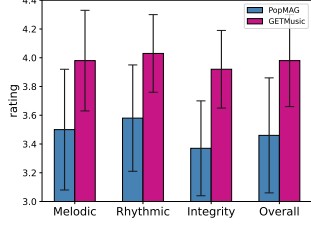

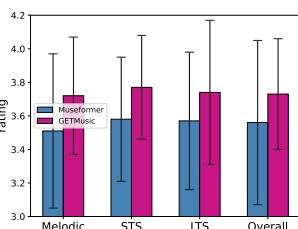

(a) Accompaniment Generation    (b) Melody Generation    (c) Generation from Scratch

Figure 4: Human evaluation ratings across three tasks.

## 5 METHOD ANALYSIS

**GETScore and GETDiff are complementary.**    To demonstrate this, we replace GETDiff with an autoregressive model. For the accompaniment generation task, we train a Transformer decoder equipped with 14 prediction heads. At each decoding step, it predicts tokens in a column of GETScore. This is denoted as GETMusic (AR). Table 2 highlights its suboptimal performance when compared to the original GETMusic. Additionally, we present the average time required in seconds for composing each musical piece using an Nvidia A100-80G GPU. Due to the considerably fewer denoising steps ($T = 100$) compared to the extensive prediction steps required by an autoregressive model, GETDiff exhibits notable speed advantages.

While it would be more informative to evaluate diffusion models trained with traditional sequence representations, this approach is intractable. Firstly, due to the inherently higher computational resource requirements of training a diffusion model compared to an autoregressive model, coupled with the fact that traditional sequence representations are usually orders of magnitude longer than GETScore when representing the same musical piece, the training cost becomes unaffordable for our source. Furthermore, diffusion models require the specification of the generation length in advance. Yet, the length of traditional sequences representing the same number of bars can vary in a wide range, leading to uncontrollable variations in the generated music's length and structure.

Based on above results and analyses, we contend that our GETScore and GETDiff together provide an efficient and effective solution for versatile and diverse symbolic music generation.

Table 2: Ablation study on generation paradigms: Autoregressive vs. Non-autoregressive.

| Method | $CA(\%)\uparrow$ | $KL_{Pitch}\downarrow$ | $KL_{Dur}\downarrow$ | $KL_{IOI}\downarrow$ | $Time\downarrow$ |
|---|---|---|---|---|---|
| PopMAG | 61.17 | 10.98 | 7.00 | 6.92 | 23.32 |
| GETMusic (AR) | 46.25 | 11.91 | 7.08 | 6.49 | 17.13 |
| **GETMusic** | **65.48** | **10.05** | **4.21** | **4.22** | **4.80** |

**Discrete diffusion models are better suited for symbolic music.**    Unlike continuous diffusion models, which require thousands of denoising steps to generate a piece of music (Mittal et al., 2021), discrete diffusion models achieve better results with just a hundred steps. As evidence of this, we trained a continuous variant (abbr. CON) of our discrete diffusion model (abbr. DIS), with hyperparameters following those detailed in (Mittal et al., 2021). The results are shown in Table 3.

Table 3: Comparing a discrete diffusion model verse a continuous diffusion model.

| Method | $CA(\%)\uparrow$ | $KL_{Pitch}\downarrow$ | $KL_{Dur}\downarrow$ | $KL_{IOI}\downarrow$ |
|---|---|---|---|---|
| Accompaniment Generation | | | | |
| CON. | 63.22 | 10.10 | 4.49 | 4.54 |
| DIS. | **65.48** | **10.05** | **4.21** | **4.22** |
| Lead Melody Generation | | | | |
| CON. | 78.51 | 10.25 | 3.80 | 3.88 |
| DIS. | **81.88** | **9.82** | **3.67** | **3.49** |
| Generation from Scratch | | | | |
| CON. | - | 8.07 | 3.63 | 5.82 |
| DIS. | - | **7.99** | **3.38** | **5.33** |

**Performance across varied lengths is stable.** In §4.3, Task 3 was tested with variable music lengths, whereas the first two tasks maintained a fixed length of 128 beats. To investigate performance across various lengths, we conducted the accompaniment generation with controlled lengths of 64 and 96 beats, respectively. Table 4 demonstrates the superior and stable performance of our method.

Table 4: Performance across varied lengths (Accompaniment Generation).

| Method | $CA\uparrow$ | $KL_{Pitch}\downarrow$ | $KL_{Dur}\downarrow$ | $KL_{IOI}\downarrow$ |
|---|---|---|---|---|
| PopMAG (64 beats) | 67.22 | 10.83 | 7.17 | 6.86 |
| **GETMusic** (64 beats) | **76.50** | **9.98** | **4.23** | **4.15** |
| PopMAG (96 beats) | 63.35 | 10.71 | 7.36 | 6.97 |
| **GETMusic** (96 beats) | **68.05** | **9.99** | **4.24** | **4.20** |

## 5.1 EXPLORING LEARNED EMBEDDINGS: A CASE STUDY ON GETSCORE

We have discovered that certain learned features within GETScore closely align with principles of music theory. For instance, although C and Cm share two same notes out of their total three, the relationship between these two chords resembles that of "gear" and "fear" in language—they seem similar at first glance but diverge significantly in semantics. C and Cm fulfill distinct roles in chord progression: while C evokes brightness, Cm evokes emotional feelings. The cosine similarity between their GETScore embeddings is measured at 0.12. Nonetheless, disparate tonalities do not inherently signify disconnection; for instance, C exhibits a stronger correlation with Am, its relative minor, boasting a similarity score of 0.43. These learned features within our GETScore closely adhere to music theories and underscore the effectiveness of GETScore.

## 5.2 EFFECTIVENESS OF CONDITION FLAGS

To evaluate the effectiveness of the condition flags, we remove them and re-train a diffusion model. We compare the new diffusion model with the original GETDiff in accompaniment generation task. We do not include the unconditional generation task in our report because the condition flags are all designated as "False," and their removal has minimal impact. The results are shown in Table 5. Given the comparable loss, removing the condition flags has minimal impact on training and convergence, but it leads to worse generation quality. This demonstrates that condition flags guide the model to generate high-quality music, particularly in conditional generation scenarios.

Table 5: Ablation study on condition flags.

| Method | $CA\uparrow$ | $KL_{Pitch}\downarrow$ | $KL_{Dur}\downarrow$ | $KL_{IOI}\downarrow$ | $Loss\downarrow$ |
|---|---|---|---|---|---|
| **GETMusic** (AG) | **65.48** | **10.05** | **4.21** | **4.22** | **1.39** |
| − condition flags | 45.16 | 10.89 | 6.32 | 5.34 | 1.40 |

## 6 CONCLUSION

We propose GETMusic, a unified representation and diffusion framework to effectively and efficiently generate desired target tracks from scratch or based on user-provided source tracks, which can address diverse composition needs. It has two core components: a novel representation GETScore and a diffusion model GETDiff. GETMusic can compose music across various source-target combinations and perform flexible zero-shot generation.

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

# A    DATA DETAILS

**Cleansing data**    We perform a data cleansing process by four steps. Firstly, we employ MIDI Miner (Guo et al., 2019) to identify the *melody* track. Secondly, we condense the remaining tracks into five instrument types: *bass*, *drum*, *guitar*, *piano*, and *string*. Thirdly, we apply filtering criteria to exclude data that contains a minimal number of notes, has less than 2 tracks, exhibits multiple tempos, or lacks the melody track. Fourthly, for all the data, we utilize the Viterbi algorithm implemented by Magenta[6] to infer the corresponding chord progression for each music piece. This serves as an additional guide for composition. Further details are discussed below. Lastly, we convert these data into GETScores and segment them into fragments of up to 512 time units.

**Chord progression**    A chord progression in music refers to a specific sequence of chords that are played or harmonized in a particular order. Chords, which are combinations of two or more notes played simultaneously, serve as the fundamental elements of harmony in music. The use of chord progressions is crucial in defining the harmony and structure of a musical composition. They play a significant role in determining the mood, emotional impact, and overall direction of the music. For instance, "C - Am - F - G" usually evokes an atmosphere of sadness and nostalgia.[7] Consequently, chord progressions are commonly used as a guiding framework in the creation of music (Zhu et al., 2018; Lv et al., 2022; Ju et al., 2021).

In GETScore, we arrange the chord progression as an extra track and it differs from regular instrumental tracks in some aspects: Although certain commonly used chords may appear in specific instrumental tracks and have been represented as pitch tokens, we do not reuse these tokens to ensure that the chord progression track provides equitable guidance for each individual track. GETMusic incorporates 12 chord roots: `C, C#, D, D#, E, F, F#, G, G#, A, A#, B` and 8 chord qualities: `major, minor, diminished, augmented, major7, minor7, dominant,` and `half-diminished`. In the chord progression track, we allocate the chord root in the first row and the quality in the second row. The chord track is entirely filled, without any paddings. Figure 5 is an illustrative configuration of GETScore used for training.

**Vocabulary**    In the last step of the cleansing process mentioned above, the construction of the vocabulary is essential before converting music fragments into GETScores. In GETScore, each track has its own pitch vocabulary, while the duration vocabulary is shared among all tracks. The maximum duration considered in this paper is 16 time units, resulting in a total of 17 duration tokens ranging from 0 (the special duration token for drums) to 16 time units. The final vocabulary consists of 17 duration tokens, 20 chord tokens, a padding token, a `[MASK]` token, an `[EMPTY]` token, and specific pitch tokens for each track: 128 for lead, 853 for bass, 4,369 for drums, 1,555 for piano, 3,568 for guitar, and 1,370 for strings. In total, the vocabulary consists of 11,883 tokens.

After implementing rigorous cleansing and filtering procedures, we produced approximately 140,000 high-quality GETScores, equivalent to approximately 3,000 hours, with the maximum $L$ as 512. We sampled 1,000 GETScores for validation, 1,000 for testing, and the rest for training.

---

[6]`https://github.com/magenta/magenta`

[7]`C, Am,` and similar symbols are chord names. To provide an illustrative example, here is a song featuring this chord progression: `https://www.youtube.com/watch?v=qiiyq2xrSI0&t=53s`

| | | | | | | | | | | | | | | | | | | | | | | | |
|---|---|---|---|---|---|---|---|---|---|---|---|---|---|---|---|---|---|---|---|---|---|---|---|
| **Melody** | 69 | | | 71 | | | 71 | | | ...... | | 80 | | | | | | | | | | | |
| | 4 | | | 4 | | | 4 | | | ...... | | 10 | | | | | | | | | | | |
| **Bass** | 13 | | | 14 | | | 19 | | | ...... | | | | | | | | | | | | | |
| | 2 | | | 2 | | | 2 | | | ...... | | | | | | | | | | | | | |
| **Drum** | 1023 | | 1247 | 1092 | | 1023 | 1247 | | | ...... | | | 1251 | | | 1251 | | | | | | | |
| | 0 | | 0 | 0 | | 0 | 0 | | | ...... | | | 0 | | | 0 | | | | | | | |
| **Guitar** | 46 | 45 | | 43 | 46 | 52 | | | | ...... | | | 65 | | | 61 | | | | | | | |
| | 1 | 2 | | 1 | 2 | 4 | | | | ...... | | | 4 | | | 4 | | | | | | | |
| **Piano** | 148 | | 153 | 101 | | 148 | 155 | 101 | | ...... | | 130 | 128 | 112 | | 136 | | | | | | | |
| | 1 | | 1 | 2 | | 1 | 1 | 3 | | ...... | | 4 | 1 | 2 | | 5 | | | | | | | |
| **String** | 792 | | | | | | | | | ...... | | 866 | | | | | | | | | | | |
| | 8 | | | | | | | | | ...... | | 8 | | | | | | | | | | | |
| **Chord** | C | C | C | C | C | C | C | C | C | C | ...... | A | A | A | A | A | A | A | A | A | A | | |
| | maj | maj | maj | maj | maj | maj | maj | maj | maj | maj | ...... | min | min | min | min | min | min | min | minj | min | min | | |

**← 512 Units →**

Figure 5: An example case showing GETScores used in our experiments. It has 6 instrumental tracks and an extra chord-progression track. During training, selected source tracks are preserved and target tracks are corrupted, as discussed in §3.2. During inference, specified source tracks are preserved, and all other parts are masked for the denoising process of GETDiff, resulting in the generation of new content.

# B  HUMAN EVALUATION GUIDELINES AND METRICS

For the accompaniment generation task, evaluators focused on several key aspects Zhu et al. (2018):

• **Melodic**: Are the musical notes harmonious?

• **Rhythmic**: Is the accompaniment fluent with appropriate pauses?

• **Integrity**: Does the structure of the accompaniment feel complete and devoid of abrupt interruptions?

• **Overall**: How well does the accompaniment integrate with the melody?

When evaluating melody generation, similar criteria were employed, with evaluators focusing on the aspects mentioned above for the melody rather than the accompaniment.

The third task, unconditional generation, requires evaluators to consider following aspects (Yu et al., 2022):

• **Melodic**: Are the musical notes harmonious?

• **Short-Term Structure (STS)**: Are there well-crafted structures in adjacent sections, such as good repetitions and effective development?

• **Long-Term Structure (LTS)**: Does the piece demonstrate cohesive structures in longer distances, like song-level repetitions?

• **Overall**: An aggregate rating of the generated music, which needs to consider melody, rhythm, and structure comprehensively.

Evaluators assigned ratings on a scale from 1 (Poor) to 5 (Excellent). Each evaluator received an hourly wage of $8, and the total evaluation process took approximately 5 hours for each group.

