# OpenReview forum: "GETMusic: Generating Music Tracks with a Unified Representation and Diffusion Framework"
_ICLR.cc/2025/Conference — ICLR 2025 Conference Withdrawn Submission_

### Official Review · Reviewer_ft4d · 2024-10-28

**Soundness:** 2
**Presentation:** 4
**Contribution:** 2
**Rating:** 6
**Confidence:** 4

**Summary:**

This paper describes a system for multi-track symbolic music generation, _GETMusic_. The authors introduce a new musical representation, _GETScore_, which compactly represents multi-track music in a two-dimensional token-based structure, and a neural model, _GETDiff_, a non-autoregressive discrete diffusion model trained to predict randomly masked tokens from symbolic music represented as a GETScore. The authors build on recent literature including piano-roll generation using diffusion models, and next-token autoregressive symbolic music generation. They evaluate their proposed system with objective musical metrics, as well as with a subjective listening test, comparing the musical quality with that of previous models. In both cases, the proposed approach performs better than the baseline.

**Strengths:**

Overall, this paper presents a generative system, which within the context of non-autoregressive generative models for symbolic music is an advancement. More specifically:

- The GETScore musical representation is well motivated, formulated, and clearly described. The reviewer deems the proposed representation as novel in the context of previous literature.

- As the authors note, GETScore is quite compact compared to the commonly used piano-roll representation. Even judged alone, GETScore is a serious contribution, and it's very possible that this representation will be useful for a variety of generative and MIR tasks, which could be useful to the community.

- The details of GETScore and GETDiff are clearly presented. Specifically, Figures 2 and 3 are very helpful to understand the intricacies of GETScore.

- There are a significant number of experiments presented in Section 4. Although systems for generative symbolic music are notoriously hard to evaluate, the authors make a significant attempt to do so as rigorously as they can. In all cases, the proposed approach performs excellently.

- The musical samples provided on the demo page are impressive, suggesting that subjectively, this framework does well at symbolic music generation.

**Weaknesses:**

- There is a potentially significant issue of missing references. The training objective and inference process for the proposed model GETDiff is quite similar in nature to those used in Huang et al. [1], in which the authors propose a discrete training objective, predicting missing notes from piano-rolls which have been randomly partially masked. At inference time, they use blocked Gibbs sampling, which is reminiscent of the inference procedure outlined in Section 3.2. Although the proposed approach is multi-track, and the framing of GETDiff as a discrete diffusion model changes the loss function, at the very least this work should be referenced and the similarities should be addressed in the related work. Some other relevant references are also missing, and are not present in ablation experiments, such as [2].

- If I understand correctly, the ablation experiment in Section 5 (L457-463) is not very well designed. By using 14 separate prediction heads and presumably sampling each column in the GETScore with a single forward pass, the training objective isn't accurately represented for GETDiff AR. As an example, when predicting the length of a note, it is impossible for the model to condition directly on the pitch of the note that it is predicting, and instead can only condition implicitly on the distribution of possible pitches predicted by the model. This introduces mathematical issues which may be responsible for the degraded performance. A much better ablation would be to compare against a transformer-decoder trained to predict the next token for a flattened version of GETScore. This should be technically possible as it would only require a context-length of 512*14=7168.

- There are some very minor issues about expressivity. According to our understanding, it is not possible to represent concurrent notes (e.g., chords) within a single track that have differing offsets.

[1] Huang, C.Z.A., Cooijmans, T., Roberts, A., Courville, A. and Eck, D., 2019. Counterpoint by convolution. arXiv preprint arXiv:1903.07227.

[2] Thickstun, J., Hall, D., Donahue, C. and Liang, P., 2023. Anticipatory music transformer. arXiv preprint arXiv:2306.08620.

**Questions:**

I would be interested in the following experiments, in addition to addressing the concerns outlined in the weakness section:

- Based on our understanding, the GETScore representation is directly compatible with the training and inference procedure used to train Coconet [1]. A comparison between these two generative systems when trained on the same dataset with the same musical representation (GETScore) would be valuable.

- Although the proposed model is quite conclusively better than the autoregressive Museformer, the problems highlighted make it unclear whether this superiority extends to autoregressive models trained faithfully on the GETScore representation (e.g., on a flattened version as described above).

---

### Official Review · Reviewer_YgLQ · 2024-10-31

**Soundness:** 2
**Presentation:** 3
**Contribution:** 2
**Rating:** 3
**Confidence:** 4

**Summary:**

The paper introduces GETMusic, a framework designed for versatile symbolic music generation that supports generating any target instrument tracks based on provided source tracks. The GETMusic framework has two main components: GETScore, a novel music representation method, and GETDiff, a diffusion-based generative model.

**Strengths:**

The paper is well-structured and clearly presented, and it addresses several important scenarios for conditional generation in symbolic music generation.

**Weaknesses:**

The biggest weakness is the limited contribution, as diffusion models for symbolic music and conditional track generation have already been explored in previous work such as AccomMontage, SongDriver. The new representation method also lacks comparisons with alternative approaches.

The experiments are incomplete; each contribution requires validation. For instance, it’s unclear how the representation method outperforms others or how the diffusion model improves over baseline diffusion models. Additionally, more recent works should be included in task-level comparisons, as PopMAG was introduced four years ago.

This also suggests that the related work survey is incomplete, omitting recent studies on conditional generation in symbolic music.

**Questions:**

Since dynamic factors like velocity and tempo variation are not considered in the paper, how does it ensure that the music sounds better than other representation methods, such as REMI, which are presented in sequence?

Is GETScore symbolic-based information? If so, why is it measured in hours?

---

### Official Review · Reviewer_iMn9 · 2024-11-02

**Soundness:** 2
**Presentation:** 3
**Contribution:** 1
**Rating:** 3
**Confidence:** 3

**Summary:**

The paper has two main contributions:

1) A symbolic music representation consisting of a tracks-by-timesteps grid, where each grid cell contains a pitch token and a duration token.  Polyphony is handled by encoding a *combination* of pitches as a single token.

2) A discrete diffusion framework that can handle arbitrary conditional generation tasks on the symbolic music grid, including unconditional generation.  For conditional tasks, the paper introduces extra flags that indicate whether each grid cell is part of the conditioning.

**Strengths:**

1) As far as I know this is the first application of discrete diffusion to symbolic music generation.

2) The generated samples sound quite good!

3) Evaluation seems good, with the caveat that I don't really trust any evaluation of generative music models :)

**Weaknesses:**

These are not in order of importance.

1) There are already more symbolic music generation representations and models out there than I can keep track of, and they all sound pretty decent.  I consider this problem basically "solved" since the release of OpenAI's MuseNet (which had no accompanying academic paper).  It's not clear that this paper is a significant advance on what is already possible.

2) The CoCoNet model by Huang et al. (https://arxiv.org/abs/1903.07227) uses a setup that is very similar to this paper: multiple tracks are generated with arbitrary segments fixed as conditioning; instead of diffusion, the remaining portions are generated iteratively using Monte Carlo sampling.

3) The paper seems confused about the taxonomy of symbolic music representations, dividing the space into "image-based" and "sequence-based" representations.  Here it would make sense to examine the pitch and time axes separately.  Either axis can be treated in dense ("image-based") or sparse ("sequence-based") fashion.

    With time, the main reason one might use a sparse approach is to handle expressive timing; the dense resolution becomes extremely high.  This paper does not model expressive timing and thus uses a dense approach, with exactly two tokens per time step.  However, it's worth noting that the approach in the paper cannot easily be extended to handle not only expressive timing, but also things like triplets, without blowing up the time dimension.

    With pitch, the main reason to use a dense approach is to handle polyphony; for monophonic music the pitch axis can be collapsed into a single value at each time.  However, for many polyphonic instruments e.g. piano, the space of possible pitches is quite large, making sparsity desirable.  This paper handles polyphony in a somewhat unique way, flattening variable-length combinations of notes into single tokens (see next item).

4) The handling of polyphony is very unsatisfying.  For example, all combinations of piano notes are compressed to a vocabulary of 1555 tokens.  This isn't even enough to represent all pairs of piano keys!  And the drum vocabulary is almost 3 times as large as the piano vocabulary; how did this end up happening?

    Here's a way polyphony could potentially have been handled that only minimally changes the setup.  On the input side, instead of blowing up the vocabulary with combinations of pitches, sum (or average) the token embeddings of all active pitches.  On the output side, instead of a softmax over pitch combination tokens, sample the binary presence/absence of each pitch independently (for a diffusion model, this independence should be okay since the other cells are sampled independently anyway) then sparsify.  This shouldn't increase memory usage since you need to construct the softmax vector anyway.

    (It's entirely possible that you already tried the above suggestion and it ended up not working; if so please disregard.)

5) I am not especially knowledgable about diffusion modeling, even less so about discrete diffusion.  But it's not clear whether the method in this paper goes beyond the standard approach.  From what I can tell, the use of condition flags is new, but that raises the question of why previous discrete diffusion methods didn't need to use such flags, and the paper provides no discussion of this.

**Questions:**

I guess my main question is: why should I use this approach for symbolic music modeling when there are so many others?  Part of the appeal of Transformer-based sequence modeling is that basically all of the work is in the training data preparation; an off-the-shelf model architecture can be used.  This seems like a lot of modeling work for not a lot of gain.

---

### Note · Authors · 2024-11-26

**Comment:**

We truly appreciate the time and effort reviewers've dedicated to this submission. We withdraw this submission to improve it based on reviewers' insightful suggestions.

**Withdrawal Confirmation:**

I have read and agree with the venue's withdrawal policy on behalf of myself and my co-authors.